# Metformin Suppresses Thioacetamide-Induced Chronic Kidney Disease in Association with the Upregulation of AMPK and Downregulation of Oxidative Stress and Inflammation as Well as Dyslipidemia and Hypertension

**DOI:** 10.3390/molecules28062756

**Published:** 2023-03-18

**Authors:** Mohammad Y. Alshahrani, Hasnaa A. Ebrahim, Saeed M. Alqahtani, Nervana M. Bayoumy, Samaa S. Kamar, Asmaa M. ShamsEldeen, Mohamed A. Haidara, Bahjat Al-Ani, Alia Albawardi

**Affiliations:** 1Department of Clinical Laboratory Sciences, College of Applied Medical Sciences, King Khalid University, Abha 61413, Saudi Arabia; 2Department of Basic Medical Sciences, College of Medicine, Princess Nourah bint Abdulrahman University, P.O. Box 84428, Riyadh 11671, Saudi Arabia; 3Department of Surgery, College of Medicine, King Khalid University, Abha 61421, Saudi Arabia; 4Department of Physiology, College of Medicine, King Saud University, Riyadh 11461, Saudi Arabia; 5Department of Histology, Kasr Al-Aini Faculty of Medicine, Cairo University, Cairo 11566, Egypt; 6Histology Department, Armed Forces College of Medicine, Cairo 11566, Egypt; 7Department of Physiology, Kasr Al-Aini Faculty of Medicine, Cairo University, Cairo 11566, Egypt; 8Department of Physiology, College of Medicine, King Khalid University, Abha 61421, Saudi Arabia; 9Department of Pathology, College of Medicine and Health Sciences, United Arab Emirates University, Al Ain 15551, United Arab Emirates

**Keywords:** thioacetamide, chronic kidney disease, renal fibrosis, TIMP-1, oxidative stress, inflammation, dyslipidemia, hypertension, metformin, AMPK

## Abstract

Toxic chemicals such as carbon tetrachloride and thioacetamide (TAA) are reported to induce hepato-nephrotoxicity. The potential protective outcome of the antidiabetic and pleiotropic drug metformin against TAA-induced chronic kidney disease in association with the modulation of AMP-activated protein kinase (AMPK), oxidative stress, inflammation, dyslipidemia, and systemic hypertension has not been investigated before. Therefore, 200 mg/kg TAA was injected (via the intraperitoneal route) in a model group of rats twice a week starting at week 3 for 8 weeks. The control rats were injected with the vehicle for the same period. The metformin-treated group received 200 mg/kg metformin daily for 10 weeks, beginning week 1, and received TAA injections with dosage and timing similar to those of the model group. All rats were culled at week 10. It was observed that TAA induced substantial renal injury, as demonstrated by significant kidney tissue damage and fibrosis, as well as augmented blood and kidney tissue levels of urea, creatinine, inflammation, oxidative stress, dyslipidemia, tissue inhibitor of metalloproteinases-1 (TIMP-1), and hypertension. TAA nephrotoxicity substantially inhibited the renal expression of phosphorylated AMPK. All these markers were significantly protected by metformin administration. In addition, a link between kidney fibrosis and these parameters was observed. Thus, metformin provides profound protection against TAA-induced kidney damage and fibrosis associated with the augmentation of the tissue protective enzyme AMPK and inhibition of oxidative stress, inflammation, the profibrogenic gene TIMP-1, dyslipidemia, and hypertension for a period of 10 weeks in rats.

## 1. Introduction

The toxic effects of certain chemicals used in industries and laboratories, such as carbon tetrachloride, mercury, and thioacetamide (TAA), on the human body are well documented [1,2]. TAA, an organosulfur compound, is a severe hepato-nephrotoxic agent that has been reported to induce (i) liver fibrosis and cirrhosis in rats associated with the augmentation of biomarkers of liver damage such as ALT, AST, gamma-glutamyl transferase, alkaline phosphatase, and bilirubin [3]; (ii) kidney damage in mice associated with the upregulation of reactive oxygen species (ROS), programmed cell death (apoptosis), and tissue collagen deposition (fibrosis) via different cell signaling pathways [4]; and (iii) liver cancer due to the increased hepatic tissue levels of ROS that progressively lead to the damage of liver DNA and the development of hepatocellular carcinoma and cholangiocarcinoma in rodent models [5]. All observed TAA-induced pathology is dependent on the exposure time of the body to this toxic compound. For example, acute liver injury developed in rats after a single injection of TAA that caused tissue necrosis and severe leukocytes infiltration after 6–60 h; however, liver tissue levels of the tissue necrosis biomarkers, inducible nitric oxide synthase (iNOS) and nuclear factor-kB (NF-kB), peaked 60 min after TAA injection [6]. Additionally, deleterious effects of TAA were observed 7 days post TAA injection, as demonstrated by liver tissue lesions, inflammatory cytokines, and activation of NF-kB as well as the decrease in hepatic antioxidant levels [7]. Furthermore, hepatic fibrosis and cirrhosis were induced in animals following TAA injections for 6 to 10 weeks [8], whereas hepatocellular carcinoma nodules appeared in mice and rats 50 to 70 weeks after TAA treatment [5].

Chronic kidney disease (CKD) imposes a great challenge to the health system because it can lead to kidney failure and is associated with high treatment costs including kidney transplantation [9,10]. Screening for CKD should be efficiently implemented to save lives and reduce the financial burden on the healthcare systems because early detection and intervention reduce the morbidity and mortality rates from CKD [10]. Kidney injury ranges from renal insufficiency to end-stage renal disease and represents kidney response to a diversity of insults such as chemicals, toxins, bacterial infections, metabolic diseases, and autoimmune diseases [11,12]. Industrial toxicants such as carbon tetrachloride and TAA are reported to induce the pathology of renal disease via oxidative stress and inflammation pathways, causing kidney tissue damage and elevation of biomarkers of kidney injury [13,14,15]. Chronic kidney disease induced by carbon tetrachloride was associated with interstitial fibrosis, glomerular damage, and infiltration of inflammatory cells [13]. TAA augmented biomarkers of inflammation and oxidative stress in kidney tissue as well as induced blood levels of urea, creatinine, and creatine kinase [14].

Metformin is a pleiotropic medicine that has multiple beneficial effects on humans and animals, in addition to its hypoglycemic uses. For example, (i) metformin helps to treat women with polycystic ovary syndrome (PCOS), which is characterized by increased insulin resistance. Metformin induced ovulation in nonobese women with PCOS better than the first-line drug for anovulatory infertility treatment, clomiphene [16]; (ii) metformin is widely used in clinical and research studies for cardiovascular and liver protection [17,18]. Metformin has been suggested to reduce body weight, inhibit the deposition of fats in blood vessels (atherosclerosis), improve hemostatic function and immune cell performance, and protect against nonalcoholic steatohepatitis-induced hepatocellular carcinoma [17,18]; (iii) metformin also prevents apoptosis and cellular deterioration with age (senescence) in intervertebral disc degeneration via autophagy stimulation through the activation of AMPK, which ameliorate the degeneration of discs in vivo [19]; and (iv) metformin ameliorates several types of kidney diseases such as autosomal dominant polycystic kidney disease and acute kidney injury; reduces mortality in patients with CKD, diabetic nephropathy [20,21], and gentamicin-induced nephrotoxicity via reducing mitochondrial ROS and hence improves mitochondrial homeostasis [22] in patients with stable chronic renal impairment [23]; and, finally, metformin was reported to decrease the risk of death in patients with kidney cancers and localized and metastatic renal cell carcinoma [24]. Therefore, this study examined if metformin can protect against TAA-induced kidney injury and fibrosis in rats using physiological and biochemical methods, basic and special histology staining, immunoblotting, and real-time PCR (to assess the relative gene expression) to test the proposed working hypothesis.

## 2. Results

### 2.1. Metformin Protects against TAA-Induced Kidney Injury

TAA is a known hepato-nephrotoxic agent [25]. We assessed kidney injury induced by TAA at the end of the experiment with and without metformin incorporation. H&E images of kidney sections (Figure 1A) prepared from the control group revealed normal histomorphology that showed renal parenchyma without pathologic changes, as demonstrated by normal Malpighian renal corpuscle with glomerular tuft capillary and narrow Bowman’s capsule, and normal proximal (Px) and distal (Ds) convoluted tubules. The tubular epithelium displayed an acidophilic cytoplasm and vesicular nuclei with a prominent nucleolus. In contrast, TAA intoxication (model group) caused kidney injury as revealed by (i) a deformed glomerular shape with wide glomerular capillaries (arrow) and Bowman’s capsule (asterisk); and (ii) tubular injury showing dilated proximal renal tubules, an attenuated epithelium (arrowheads), apical cytoplasmic loss, reduced brush border, necrotic luminal debris, and interstitial edema. Metformin treatment (treated group) substantially but not completely protected against TAA-induced kidney injury. It displayed partial normalization of histomorphology and regenerative changes in tubules, and some Malpighian renal corpuscles showed dilated glomerular capillaries (arrow). Quantitative analysis of the Bowman’s spaces (also known as urinary space or capsular space) of Bowman’s capsule showed a significant (*p* < 0.0001) reduction in the dilatation of the Bowman’s spaces (Figure 1B). TAA also augmented blood levels of the kidney injury biomarkers urea (Figure 1C) and creatinine (Figure 1D), which were significantly (*p* < 0.0001) decreased by metformin treatment (Met + TAA) to values still significant (*p* ≤ 0.0073) compared with those of the control rats.

### 2.2. Metformin Protects against TAA-Modulated Kidney Levels of AMPK, Oxidative Stress, and Inflammation

Oxidative stress and inflammation are well-known processes involved in the pathogenesis of kidney injury [26], and the mechanism of metformin action is reported to occur via activating the tissue-protective enzyme AMPK [27]. In light of our findings of TAA-induced kidney damage that was protected by metformin, we assessed the kidney tissue levels of phospho-AMPK (Figure 2A,B), total AMPK (Figure 2A,C), MDA (Figure 2D), SOD (Figure 2E), and TNF-α (Figure 2F) as well as blood levels of MDA, hsCRP, and TNF-α (data not shown) in all rats at the end of the experiment. Compared with the control rats, TAA caused a profound decrease in the expression of the active form of the enzyme AMPK (p-AMPK, but not the total AMPK) and SOD, and augmentation of MDA, TNF-α, IL-6 (data not shown), and hsCRP (data not shown) levels, which were effectively but not completely protected by metformin (Met + TAA).

### 2.3. Metformin Is Associated with the Protection against Kidney Fibrosis Induced by TAA

We tested the hypothesis that metformin can protect against TAA-induced kidney fibrosis. The levels of TIMP-1 gene expression (the profibrogenic biomarker) and collagen fiber deposition (fibrosis) in kidney tissues were assessed at the end of the experiment in all animal groups using a quantitative real-time polymerase chain reaction (qPCR) technique and a special histological staining method with Sirius red (Figure 3). The model group of rats (TAA) showed an increase in TIMP-1 relative gene expression, which was effectively (*p* < 0.0001) inhibited by metformin (Figure 3A). Compared with a minimal collagen staining in kidney sections prepared from the control rats (Figure 3B) that revealed a fine deposition of collagen in the basement membrane of the renal corpuscle (arrow) and the convoluted tubules (arrowhead), kidney tissue sections of the TAA-intoxicated rats (Figure 3C) depicted substantially thick collagen deposition in the basement membranes (arrow), in the interstitium (star), and around blood vessels (Bv). Metformin treatment (Met + TAA) of the experimental group for 10 weeks significantly (*p* < 0.0001) but not completely protected against TAA-induced fibrosis (Figure 3D–F).

### 2.4. Metformin Protects against TAA-Induced Dyslipidemia and Systemic Arterial Pressure

TAA is known to induce lipid dysregulation [28,29], dyslipidemia, and fibrosis-induced hypertension [29,30]. Therefore, the extent of the inhibition of dyslipidemia and hypertension by metformin against TAA-modulated blood lipids and systemic arterial pressure was evaluated in all rats at the end of the experiment. TAA caused a substantial increase in the blood levels of TG (Figure 4A), cholesterol (Figure 4B), vLDL-C (Figure 4C); systolic blood pressure (SBP) (Figure 4E); and diastolic blood pressure (DBP) (Figure 4F), which were significantly (*p* ≤ 0.0077) decreased by metformin treatment (Met + TAA) to levels still significant (*p* ≤ 0.0217) compared with those of the control group of rats for all these parameters. This meant a partial inhibition by metformin. On the other hand, metformin treatment (Met + TAA) significantly (*p* < 0.0001) augmented the HDL-C blood levels that were ameliorated by TAA (Figure 4D).

### 2.5. Correlation between Kidney Fibrosis Score and Biomarkers of Kidney Injury

We determined the correlation between kidney fibrosis score and phosphorylated AMPK, oxidative stress (MDA), inflammation (TNF-α), renal injury biomarkers (urea and creatinine), dyslipidemia (TG), and systemic hypertension (SBP). This links the pathology of TAA kidney intoxication with the biomarkers of kidney injury, and it further supports the pleiotropic effects of metformin. The kidney fibrosis score displayed a significant (*p* < 0.0001) negative correlation with phosphorylated AMPK (r = −0.975) (Figure 5A, and a positive correlation with MDA (r = 0.955) (Figure 5B), TNF-α (r = 0.954) (Figure 5C), kidney tissue injury (assessed as increase in Bowman’s capsule space) (r = 0.963) (Figure 5D), urea (r = 0.874) (Figure 5E), creatinine (r = 0.922) (Figure 5F), TG (r = 0.832) (Figure 5G), and SBP (r = 0.905) (Figure 5H).

## 3. Discussion

This article investigated the induction of chronic kidney disease (CKD) with the hepato-nephrotoxic agent TAA with and without the incorporation of the antidiabetic drug metformin in a rat model of the disease. We modeled this disease to test the hypothesis that metformin can ameliorate kidney injury and fibrosis induced by TAA, associated with the augmentation of phosphorylated AMPK and inhibition of biomarkers of inflammation, oxidative stress, dyslipidemia, and hypertension (Figure 6). Here, we report that the induction of CKD by TAA caused, after 10 weeks, profound renal cortical kidney damage and fibrosis in kidney tissues harvested from the model group of rats. This was associated with the inhibition of kidney tissue levels of phosphorylated AMPK and the antioxidant SOD and augmentation of the oxidative stress biomarker (MDA), inflammation biomarkers (TNF-α), TIMP-1, urea, creatinine, and lipid profile, as well as systemic arterial pressure, which appeared to be protected by metformin (Figure 1, Figure 2, Figure 3 and Figure 4). In addition, using the data obtained from the three animal groups, a significant correlation was observed between the renal fibrosis score and CKD biomarkers (Figure 5), which further confirms that metformin is a beneficial pleiotropic medicine to treat TAA-induced CKD. Therefore, our data support our working hypothesis mentioned above.

Elevated levels of kidney injury biomarkers such as inflammation, oxidative stress, urea, and creatinine are reported in many kidney diseases such as diabetic nephropathy [31,32], renal vasculitis [33], nephrotoxicity induced by carbon tetrachloride [15], nephrotoxicity induced by paracetamol overdose [26], and kidney injury induced by TAA (0.3% dissolved in drinking water) given to baby rats (4 weeks old; 70–80 g weight) for a period of two weeks [14]. These reports are in agreement with our data presented in this study, as shown in Figure 1 and Figure 2. In addition, our data that point to the induction of renal cortical injury in rats by TAA after 10 weeks, depicted in Figure 1, are in agreement with those obtained from previous work on carbon-tetrachloride- or TAA-induced glomeruli changes with endothelial cell swelling in rats after 3 months [34]. Furthermore, our data that point to the inhibition of kidney damage and fibrosis associated with the upregulation of phosphorylated AMPK and inhibition of biomarkers of kidney injury, inflammation, oxidative stress, TIMP-1, and hyperlipidemia by metformin are congruous with those of previous reports [35,36], which demonstrated that (i) metformin inhibited stroke damage induced by transient occlusion to the middle cerebral artery in nondiabetic mice with chronic kidney disease associated with the upregulation of AMPK and downregulation of apoptosis, as well as increased neurone survival; (ii) metformin ameliorated acute kidney injury and chronic kidney disease; (iii) metformin decreased renal cell carcinoma, renal fibrosis, podocyte loss and apoptosis of mesangial cells, and protected renal tubular cells from the adverse effects of the inflammation; and (iv) the protective property of metformin in kidney disease is associated with the activation of the AMPK cell signaling and other pathways such as inhibiting the oxidative stress, endoplasmic reticulum stress, inflammation, lipotoxicity, and antifibrotic effects. However, clinical trial studies showed that metformin is associated with lactic acidosis when the estimated glomerular filtration rate (eGFR) reached below 30 mL/min/1.73 m^2^, and this may lead to renal imperilment, which requires discontinued treatment with metformin [37,38].

Lowered blood pressure by metformin in nondiabetic models was previously reported in (i) a mouse model of angiotensin-II-induced hypertension treated daily with metformin (300 mg/kg body weight) in association with the augmentation of p-AMPK and phospho-endothelial nitric oxide synthase (p-eNOS) and inhibition of the oxidative stress enzyme nicotinamide adenine dinucleotide phosphate (NADPH) oxidase in mesenteric resistance arteries [39]; (ii) spontaneously hypertensive rats treated with 350 to 500 mg/kg per day metformin; however, metformin treatment did not affect the control normotensive WKY rats [40]; (iii) nondiabetic patients with obesity or with impaired glucose tolerance gathered from 4113 participants in a meta-analysis of randomized controlled trials [41]; (iv) a mouse model of pre-eclampsia induced by a high-fat diet in pregnant mice treated daily with metformin (20 mg/kg). Metformin is also associated with the inhibition of proteinuria and improved fetal and placental weights [42]. (v) A rat model of carbon tetrachloride-induced liver cirrhosis or common bile duct ligation-induced liver cirrhosis treated daily with metformin (300 mg/kg) caused a decrease in portal pressure and hepatic vascular resistance as well as inhibition of liver fibrosis, profibrosis biomarker alpha-smooth muscle actin (α-SMA), hepatic inflammation, and oxidative and nitrosative stress [43]. These reports are in agreement with our data shown in Figure 1, Figure 2, Figure 3 and Figure 4.

In summary, our data demonstrate that the induction of CKD associated with the modulation of p-AMPK, inflammation, oxidative stress, profibrosis and fibrosis, dyslipidemia, and systemic hypertension by the hepato-renal toxic compound TAA appear to be protected by metformin for a period of 10 weeks in rats. In addition, we demonstrated a link between renal fibrosis and the above-mentioned parameters, with metformin showing useful renal pleiotropic effects. Thus, these findings may be translated into clinical therapy.

## 4. Materials and Methods

### 4.1. Animals

Rats (albino male rats; 170–200 g) were kept in a clean animal room inside an animal facility with a controlled temperature of 22 ± 2 °C and 50 ± 10% relative humidity. They were held in cages with 12 h light/dark cycles and permitted unrestricted access to water and food. All experiments were accomplished in accordance with the agreed guidelines mentioned in detail in the Institutional Review Board Statement section.

### 4.2. Experimental Design

A total of 24 rats were divided equally into three groups (*n* = 8 rats per group) after a one-week acclimatization period. To induce kidney injury via TAA intoxication, the model group of rats (TAA) was injected with TAA (200 mg/kg, i.p.) twice a week for 8 weeks [8], whereas control rats were injected with the vehicle. To assess the effects of metformin on TAA-induced kidney damage, a second cohort of rats (Met + TAA) was pretreated with 200 mg/kg metformin from the first day, until being culled in week 10. This group received TAA injections similar to the model group from week 3 to week 10. Rats were then anesthetized at the end of the experiment using sodium thiopentone at 40 mg/kg body weight, and blood was collected using cardiac puncture into plain tubes for serum separation and sodium-citrate-containing tubes for plasma separation. Following that, animals were sacrificed and kidneys were harvested.

### 4.3. Measurements of hsCRP, ALT, TNF-α, MDA, Urea, Creatinine, Triglyceride, Cholesterol, vLDL-C, and HDL-C

ELISA kits for the determination of blood and kidney tissue levels of highly sensitive C-reactive protein (hsCRP, ASSAYPRO, St. Charles, MO, USA), ALT (Randox Laboratories, Crumlin, UK), tumor necrosis factor-alpha (TNF-α, Abcam, Cambridge, UK), and lipid peroxidation measured as malondialdehyde (MDA, Cyaman Chemical, Ann Arbor, MI, USA) were used according to the manufacturer’s instructions. Blood urea and creatinine were measured using colorimetric methods (BioAssay System, Hayward, CA, USA). Serum levels of triglycerides (TGs), cholesterol, very-low-density lipoprotein (vLDL-C), and high-density lipoprotein cholesterol (HDL-C) were measured using commercial kits supplied by SPINREACT, Coloma, Girona, Spain.

### 4.4. Determination of Arterial Blood Pressure

Systolic and diastolic blood pressure (SBP and DBP, respectively) were measured in conscious rats using a BP monitor (LE 5001, LETICIA scientific Instruments, Barcelona, Spain), as previously reported [29]. The animals involved in the study were warmed at 28 °C for half an hour in a heating cabinet (Ugo Basile, Gemonio, VA, Italy), a process that helped the detection of the tail artery pulse. Through a miniaturized cuff, the tail was passed, and a tail-cuff sensor was attached to an amplifier (LE 5001, LETICIA scientific Instruments, Barcelona, Spain). The cuff was joined to a tail-cuff sphygmomanometer, blood pressure was recorded on a chart, and the average of three readings was taken.

### 4.5. Histological Analysis

Kidney tissues were fixed in formal saline (10%) and paraffin blocks were prepared using standard procedure. Sections of 5 μm thickness were stained with hematoxylin and eosin (H&E) for basic staining histological analysis [44]. To quantify the deposition of collagen fibers in kidney tissues, Sirius red staining was performed. Following dewaxing and rehydration of kidney tissue sections, slides were incubated with 0.1% Sirius red (Sigma-Aldrich, Gillingham, UK) overnight, dipped in 0.01 M hydrochloric acid, and dehydrated with serial ethanol concentrations without water. Determination of collagen deposition (area percentage) in sections stained with Sirius red and Bowman’s capsule space (μm) was evaluated in 10 nonoverlapping fields for each group using a Leica Qwin 500 C image analyzer (Cambridge, UK). Data were summarized as means ± SD and compared using ANOVAs followed by a Tukey test. *p*-values < 0.05 were considered statistically significant. Calculations were made using SPSS software, version 19.

### 4.6. AMPK Western Blotting Analysis

Proteins (20 μg per sample) extracted from kidney tissues were immunoblotted as mentioned previously [45]. Membranes were probed at 4 °C overnight with the primary antibodies anti-AMPK-phospho-Thr172, anti-AMPK, and beta-actin (Cell Signaling Technology, Danvers, MA, USA). Proteins were made visible with an ECL detection kit (Merck Life Science, Gillingham, Dorset, UK). Relative expression was resolved using image analysis software to obtain the intensity of the target protein bands with regard to a control sample after normalization on Chemi Doc MP imager by β-actin.

### 4.7. Kidney Tissue Inhibitor of Metalloproteinases-1(TIMP-1) Gene Expression Using Real-Time PCR (qPCR)

Total RNA was prepared from the kidney tissue of rats using an RNeasy Mini Kit (Qiagen Pty, Victoria, Australia), and the RNA (1 μg) was reverse-transcribed with a complementary DNA (cDNA) synthesis kit (Fermentas, Waltham, MA, USA). cDNA samples and standards (triplicates) were amplified in Master Mix containing SYBR green (Thermo Fisher Scientific Inc., Waltham, MA, USA) with primers specific for TIMP-1(sense, 5′-GGT TCC CTG GCA TAA TCT GA-3′; antisense, 5′-GTC ATC GAG ACC CCA AGG TA-3′) or the housekeeping gene, β-actin, using the standard qPCR technique [39].

### 4.8. Statistical Analysis

The data are expressed as mean ± standard deviation (SD). Data were processed and analyzed using the SPSS version 10.0 (SPSS, Inc., Chicago, IL, USA). One-way ANOVA was performed followed by Tukey’s post hoc test. Pearson correlation statistical analysis was performed for the detection of a probable significance between two different parameters. Results were considered significant if *p* ≤ 0.05.

## Figures and Tables

**Figure 1 molecules-28-02756-f001:**
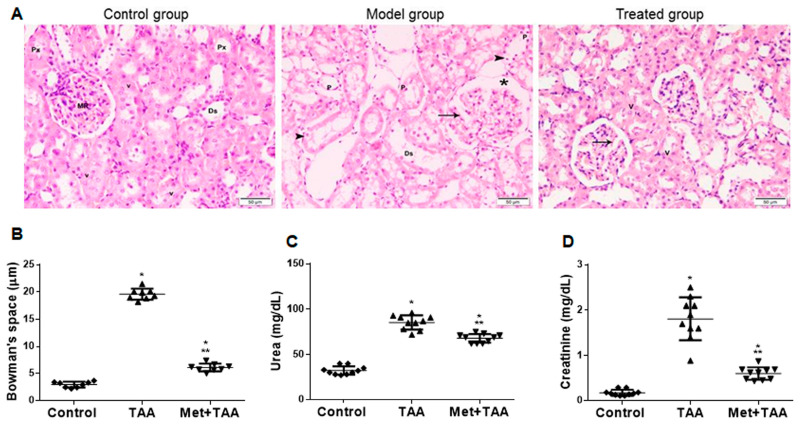
Induction of kidney injury by TAA is inhibited by metformin (Met). (**A**) H&E-stained images (×200) of kidney sections obtained at the end of the experiment, after 10 weeks, from the control, model (TAA), and treated (Met + TAA) groups of rats are shown. Note that arrows point to dilated glomerular capillaries, whereas arrowheads point to dilated proximal renal tubules. The asterisk points to dilatation of the Bowman’s space. Px: proximal convoluted tubules; Dx: distal convoluted tubules; MR: Malpighian renal corpuscle; p: pyknotic nuclei; v: vesicular nuclei. The scatterplots in (**B**) represent a quantitative analysis of the Bowman’s capsule space in kidney sections from the rats’ groups; Control (
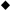
), TAA (
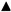
), and Met + TAA (
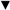
). Blood levels of urea (**C**) and creatinine (**D**) were measured at the end of the experiment, week 10. All of the *p* values shown are significant. * *p* ≤ 0.0073 versus control; ** *p* < 0.0001 versus TAA. *n* = 8 rats per group. TAA: thioacetamide.

**Figure 2 molecules-28-02756-f002:**
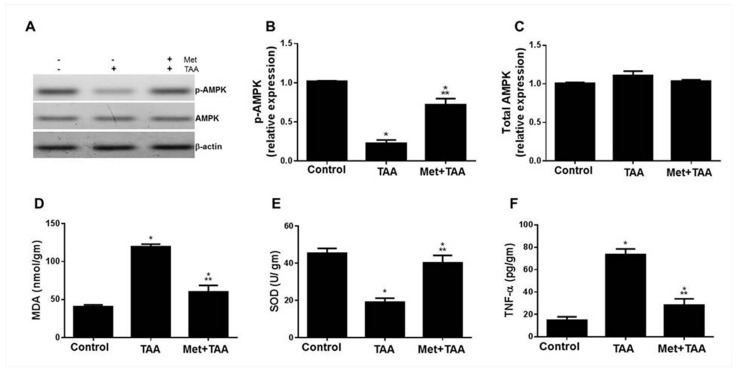
Modulation of AMPK, oxidative stress, and inflammation by TAA is protected by metformin (Met). Kidney lysates prepared from all rat groups at the end of the experiment were immunoblotted with anti-p-AMPK, anti-total AMPK, and anti-β-actin as a housekeeping control protein (**A**). The relative protein expression of these parameters is shown in (**B**,**C**). (**D**–**F**) Tissue levels of MDA (**D**), SOD (**E**), and TNF-α (**F**) were measured at the end of the experiment, week 10. In (A), − −, represent the Control; − +, represent TAA group; and + +, represent the treated group (Met + TAA). All of the p values shown are significant. * *p* ≤ 0.0073 versus control; ** *p* < 0.0001 versus TAA. *n* = 8 rats per group. AMPK: AMP-activated protein kinase; MDA: malondialdehyde; SOD: superoxide dismutase; TNF-α: tumor necrosis factor alpha; TAA: thioacetamide.

**Figure 3 molecules-28-02756-f003:**
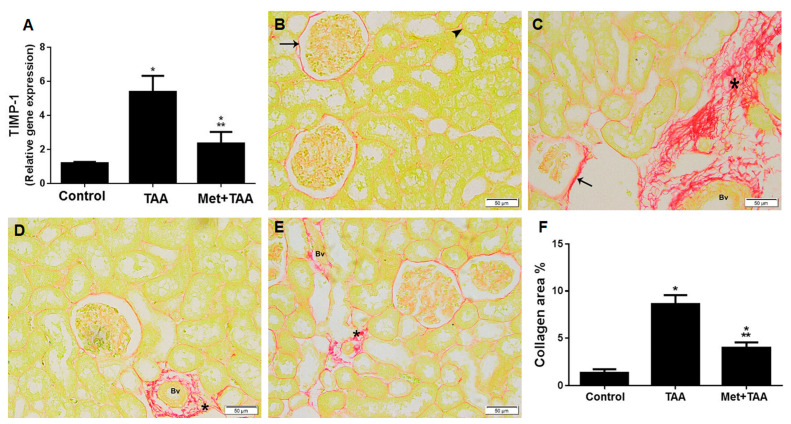
Metformin (Met) ameliorates TAA-induced kidney fibrosis. (**A**) TIMP-1 relative gene expression was assessed in all rats at week 10. Sirius-red-stained images (×200) of kidney sections from the control (**B**), TAA (**C**), and Met + TAA (**D**,**E**) groups of rats are shown. Note that arrows point to collagen deposition in the basement membranes, whereas stars point to collagen deposition in the interstitium. The arrowhead points to the convoluted tubules. Bv: blood vessels. The bar graph in (**F**) represents a quantitative analysis of the fibrosis deduced from Sirius red stain. All of the *p* values shown are significant. * *p* < 0.0001 versus control; ** *p* < 0.0001 versus TAA. *n* = 8 rats per group. TIMP-1: tissue inhibitor of metalloproteinases-1; TAA: thioacetamide.

**Figure 4 molecules-28-02756-f004:**
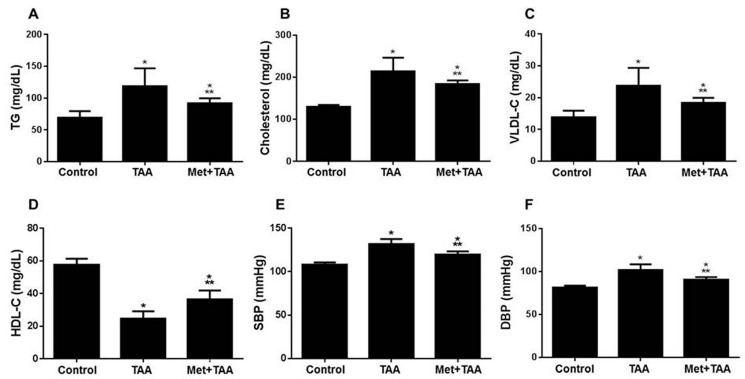
Metformin (Met) inhibits TAA-induced dyslipidemia and hypertension. Blood levels of TG (**A**), cholesterol (**B**), vLDL-C (**C**), and HDL-C (**D**) were assessed after 10 weeks in all rats. (**E**,**F**) Non-invasive blood pressure, SBP (**E**) and DBP (**F**) were measured at the end of the experiment in week 10 for all rats. All of the *p* values shown are significant. * *p* ≤ 0.0217 versus control; ** *p* ≤ 0.0077 versus TAA. *n* = 8 per group. TG: triglycerides; vLDL-ch: very-low-density lipoprotein cholesterol; HDL-ch: high-density lipoprotein cholesterol; SBP: systolic blood pressure; DBP: diastolic blood pressure; TAA: thioacetamide.

**Figure 5 molecules-28-02756-f005:**
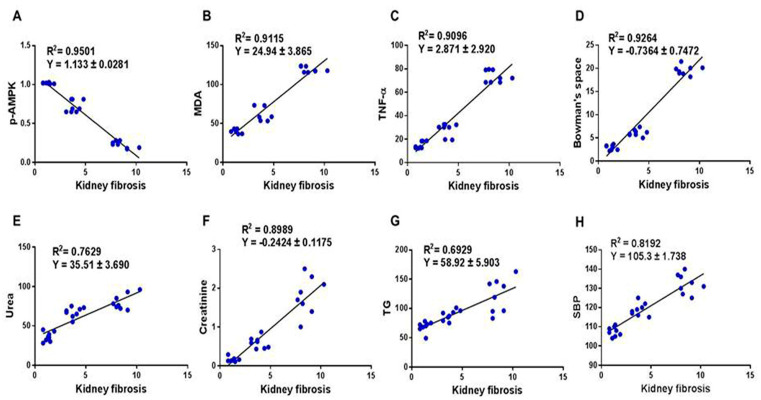
Correlation between kidney fibrosis score and phosphorylated AMPK, oxidative stress, inflammation, kidney injury, and hypertension. All rats (*n* = 8 per group) had their collagen deposition (fibrosis) in kidney tissues assessed after 10 weeks, and the relationship between fibrosis and p-AMPK (**A**), MDA (**B**), TNF-α (**C**), Bowman’s space (**D**), urea (**E**), creatinine (**F**), TG (**G**), and SBP (**H**) is shown. AMPK: AMP-activated protein kinase; MDA: malondialdehyde; TNF-α: tumor necrosis factor alpha; TG: triglyceride; SBP: systolic blood pressure.

**Figure 6 molecules-28-02756-f006:**
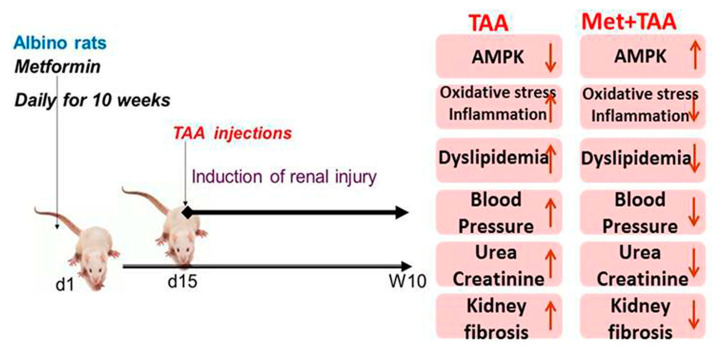
Proposed model for TAA-induced kidney injury and fibrosis, which appear to be ameliorated by metformin. TAA: thioacetamide; Met: metformin; AMPK: phosphorylated AMP-activated protein kinase; d1 and d15: day 1 and day 15, respectively; W10: week 10.

## Data Availability

The data that support the findings of this study are available on request from the corresponding author.

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
