# Peer review of "Metformin Suppresses Thioacetamide-Induced Chronic Kidney Disease in Association with the Upregulation of AMPK and Downregulation of Oxidative Stress and Inflammation as Well as Dyslipidemia and Hypertension"

_molecules, 2023, doi:10.3390/molecules28062756_

Round 1
Reviewer 1 Report (Previous Reviewer 1)
No further comments.
Author Response
We thank the reviewer for having no further comments.
Reviewer 2 Report (Previous Reviewer 3)
The authors have improved the manuscript to some extent. But the images for H&E staining in Figure 1 and Sirius red stain in figure 3 need improvements; or please provide images with high magnification.
Author Response
We thank the reviewer for his/her comment. As suggested, we have improved the resolution of Figures 1 and 3 (JPEG image with 600 dpi) and included in the revised manuscript.
Round 2
Reviewer 2 Report (Previous Reviewer 3)
I don't have further concerns.
This manuscript is a resubmission of an earlier submission. The following is a list of the peer review reports and author responses from that submission.
Round 1
Reviewer 1 Report
The manuscript entitled “Metformin Suppresses Thioacetamide-Induced Chronic Kidney Disease in Association with the Inhibition of Kidney Fibrosis and Dyslipidemia and Hypertension” was reviewed” evaluated the therapeutic potential of Metformin on thioacetamide-induced chronic kidney disease in association with the inhibition of renal fibrosis, dyslipidemia, and systemic hypertension, as well as blood levels of MDA/TNF-α/CRP, in a rodent model. The study is interesting and contributes to knowledge in the field. However, some points deserve further attention to strengthen the manuscript.
Major comments
1) To further substantiate the findings, provide the analyses of inflammation and oxidative stress in kidney samples.
2) For the sake of transparency, all bar charts should be changed to bar chart plus scatter chart format. Please indicate the number of animals used in each experiment. Were male or female rats used?
3) How many images were evaluated for Sirius Red and collagen deposition in each animal? And how many glomeruli were analyzed?
4) Is there a score of kidney injury available for figure 2D?
Reviewer 2 Report
This manuscript reports that metformin alleviates thioacetamide-induced renal fibrosis by inhibiting dyslipidemia and hypertension. Since chronic kidney disease and renal fibrosis is one of the significant issues in internal medicine and related biology, and metformin is a famous drug that is used for diabetes, the subject of this study will be of great interest to the readers of molecules and researchers as well as physicians in the related field. However, this study is at the level of preliminary experiments, and thus, the conclusion is immature yet, just showing simple phenomenal results.
The mode of action of metformin has been identified at the molecular level, and the readers of molecules are eager to know the molecular mechanism involved in bioactive molecules-exerted functions. Therefore, it is strongly recommended to include the mechanism underlying the alleviating effect of metformin on TAA-induced CKD and fibrosis.
The result of figure 5 doesn't seem to be needed in this study and should be deleted.
Reviewer 3 Report
In the manuscript, the authors demonstrate that metformin provides profound protection against TAA-induced kidney damage and fibrosis and biomarkers of renal injury associated with the inhibition of dyslipidemia and hypertension in rats. This work is not novel enough. I have several important concerns for this manuscript.
Major concerns:
1. It is not appropriate to draw the conclusion that metformin suppresses Thioacetamide-induced chronic kidney disease which is associated with the inhibition of kidney fibrosis and dyslipidemia and hypertension;
2. The topic of this manuscript is not novel enough. Most importantly, there is no novelty in mechanism underlying how metformin provides protective role against TAA-induced kidney damage;
3. Please show the n value for Figure 1-4;
4. The images for H&E staining in Figure 1 and 2 need improvements; or please provide images with high magnification.